# First Experiences with Last Aid Courses as Tool for Public Palliative Care Education in Brazil

**DOI:** 10.3390/nursrep15110386

**Published:** 2025-10-31

**Authors:** Karin Schmid, Patricia Maluf Cury, Marina Schmidt, Georg Bollig, Janaina Santos Nascimento

**Affiliations:** 1Last Aid Brazil, São Paulo 04631-011, SP, Brazil; 2Last Aid Research Group International (LARGI), 24837 Schleswig, Germany; georg.bollig@uk-koeln.de; 3FACERES Medical School, São José do Rio Preto 15090-305, RJ, Brazil; 4Hospital de Base, São José do Rio Preto 15090-000, SP, Brazil; 5Last Aid Germany, 24837 Schleswig, Germany; 6Department of Palliative Medicine, Faculty of Medicine and University Hospital, University of Cologne, 50931 Cologne, Germany; 7Department of Anesthesiology, Intensive Care, Palliative Medicine and Pain Therapy, Helios Klinikum, 24837 Schleswig, Germany; 8Department of Occupational Therapy, Federal University of Rio de Janeiro, Rio de Janeiro 21941-853, RJ, Brazil; jananascimento@medicina.ufrj.br

**Keywords:** palliative care, compassionate communities, last aid course, public palliative care education, Brazil, end of life care

## Abstract

**Background/Objectives:** Promoting access to palliative care education at all levels and in diverse contexts is essential. In Brazil, however, despite progress, awareness remains limited. The Last Aid approach provides accessible ways for the public to engage in discussions about serious illness, death, dying, and grief, while also suggesting practical actions to support. The present study aimed to investigate whether Last Aid Courses are accepted and contribute to increasing knowledge and awareness of Palliative Care to different settings in Brazil. **Design/Methods**: To obtain more in-depth views, a mixed methods approach was chosen, and participants from all Last Aid Courses offered in Brazil between March and November 2024 were invited to respond to a mixed qualitative–quantitative questionnaire provided after the course. **Results:** Thirty-two courses were offered, with 343 participants. Most of the Last Aid Courses participants came from the general public (53.2%), followed by health students (28.1%). 98.8% of all respondents indicated they had acquired new knowledge. Qualitative analysis identified four themes: death as part of life, communication about dying, dignity and respect for patients’ wishes, and the need for palliative care education. Participants highlighted autonomy, compassion, and dialogue as essential, reinforcing the urgency of expanding public education. **Conclusions**: The course implementation in Brazil showed positive results, indicating its potential to raise awareness about the topic, regardless of the context.

## 1. Introduction

Palliative care is a specialized form of medical care designed to provide relief from the symptoms, pain, and emotional distress associated with serious illnesses. It focuses on improving the quality of life for patients and their families and is applicable from the moment of diagnosis through the progression of the illness and up until the end of life [1,2,3,4]. Although significant efforts are directed toward curative treatments in several countries, efforts focused on palliative care and end-of-life care remain limited. Furthermore, important disparities exist in the quality of palliative care services across different countries [1] leaving many seriously ill and dying people in need without adequate palliative and compassionate care around the end-of-life.

To better illustrate these disparities, the results of the latest ranking published by The Economist on the quality of death in various countries showed the U.K., Australia, New Zealand, Ireland, Belgium, Taiwan, Germany, the Netherlands, the U.S. and France among the Top Ten. Brazil ranked 42nd among the 80 countries analyzed. To prepare this ranking, 20 quantitative and qualitative indicators were analyzed and categorized into five categories: the palliative and health environment, human resources, accessibility to care, care quality, and the level of community engagement. Furthermore, Brazil ranked third to last in terms of quality of death among 81 countries studied, ahead only of Lebanon and Paraguay, according to the international report published in the Journal of Pain and Symptom Management [1,5].

These findings reveal not only structural inequalities but also cultural barriers. One possible explanation for this result is linked to a widespread culture of denying the reality of death and dying [2]. Thus, it is crucial to foster cultural changes that acknowledge the significance of palliative care, recognize death as a natural aspect of life, and emphasize the quality of life for patients throughout their entire life journey, including serious illness and end of life, for both patients and their families [5].

In this context, Brazil has made gradual progress, marked by the significant achievement of the National Palliative Care Policy published in 2024 [6]. This policy aims to enhance autonomy and quality of life for patients, focusing on symptom control to relieve suffering, the development of advanced care plans, and end of life care in the community. One major point of the policy is to promote awareness and education about palliative care throughout society [6]. Unfortunately, at present palliative care is widely unknown to the general public as well as to health professionals in Brazil [7].

Given this scenario, education emerges as a key driver for change. Education is fundamental in palliative care; it should be accessible at all levels and tailored to various contexts, including public palliative care education (PPCE) [8,9]. This is particularly important in a country like Brazil, where some regions still face challenges in accessing essential services that contribute to quality of life, including healthcare, food, housing, income, education, and leisure [8,9,10] and adapts to different realities [4].

Among the innovative educational strategies developed, the Last Aid Course is particularly noteworthy. The concept of Last Aid is founded on the idea that knowledge of palliative care should be included in public education as part of a so-called public knowledge approach [11,12]. The aim is to cultivate compassionate communities and to transform end-of-life care into a collective responsibility [11,12]. The Last Aid approach provides accessible ways for the general public to engage in discussions about serious illness, dying, death and grief, and it also suggests practical Last Aid measures to offer support in various contexts [12].

The origins of this movement further highlight its global scope. The concept was first described by Georg Bollig in 2008, and courses started in 2015 in Germany, Norway, and Denmark. Through a standardized curriculum and a slide set, the courses are held in 23 countries, including Germany, Australia, Scotland, Slovenia, Brazil, and Singapore [12,13,14,15].

In 2020, Last Aid Courses were started in Brazil, utilizing an international curriculum and slide set. Since then, Last Aid Brazil has offered both face-to-face and online courses in various locations, including cultural centers, healthcare centers, nursing homes, universities, and compassionate communities [16]. The course usually lasts 4 h (=240 min), including breaks.

Evidence from previous studies reinforces the relevance of this initiative. Research findings indicate that the Last Aid Course is not only feasible and well received but also has significant potential to enhance palliative care education for the general public across various countries [11,13,17], including children and adolescents [18]. Brazil is the first country in South America that has started to implement Last Aid Courses. However, despite this progress, a scientific evaluation of Last Aid Courses in Brazil and South America is still needed to evaluate the feasibility and acceptance in this part of the world with huge geographical distances and different ethnicities. Considering this, it is crucial to conduct a study on Last Aid Courses in Brazil to better understand its potential, challenges, and the experiences involved in its implementation across the country. This investigation is particularly relevant, as emphasized by Bollig and Bauer as research indicates that outcomes may vary depending on the location where Last Aid Courses are implemented [19]. Furthermore, the Last Aid Course addresses a new demand arising from the National Palliative Care Policy in Brazil [6].

In addition, Brazil faces unique challenges, such as regional disparities in access to healthcare services. There is a need for better training of healthcare professionals, raising public awareness about palliative care, recognizing it as an integral part of patient care, and improving access to medications for pain relief and symptom management [20].

This study is also aligned with the needs outlined in the Global Atlas of Palliative Care, which highlighted the importance of strategies to promote education in palliative care and research on the topic, especially in low- and middle-income countries [21].

Building on these considerations, this study aims to investigate if Last Aid Courses can contribute to bringing knowledge and awareness of Palliative Care to different settings in Brazil. We also evaluated participants’ satisfaction, perceptions of program quality, and willingness to recommend the course, with the purpose of offering a more comprehensive view of its educational and social impact.

## 2. Materials and Methods

### 2.1. Study Design

The research design is a cross-sectional study, collecting data of the participants after approval from the ethics committee. It is based on a mixed-methods approach with a combination of quantitative and qualitative data from a questionnaire (Appendix A). Being the first Brazilian course evaluation, the mixed methods approach was chosen to provide a richer picture, with more in-depth views of the participants. It employed a descriptive analysis design to investigate the frequency of the quantitative data collected, and a qualitative research design to explore the perceptions and experiences of the participants. The analysis and presentation of the qualitative data was based on qualitative description [22,23].

### 2.2. Participants, Data Collection and Analysis

Between March and November 2024, all Last Aid Course participants were invited to participate in the study. All were given a brief explanation of the importance of evaluating the course to improve its efficiency, quality, and outcome. They were informed that participation in the study was voluntary. After providing informed consent to participate in the study, the participants were asked to answer a questionnaire lasting approximately 10 min, virtually, using a google forms and included the informed consent statement (Appendix B).

The present study is a part of an ongoing larger research project that aims to analyze the understanding and impact of the Last Aid Course on the perception of finitude, care, mourning, death literacy, and breaking of taboos among the participants in Brazil.

The inclusion and exclusion criteria were as follows:Inclusion criteria: all questionnaires from Last Aid Course participants from 18 years of age, who provided informed consent and completed all the answers to the questionnaires properly.Exclusion criteria: participants who did not respond to the questionnaire.

The data collection was conducted virtually through a web-based questionnaire created on the Google Forms platform, featuring both open and closed-ended questions. Data analysis regarding participant characteristics was performed using absolute and relative frequencies. The open-ended questions were grouped into thematic categories and counted after their inductive evaluation.

In order to protect the participants’ privacy, no personal data other than age, sex, and profession were collected. Participants could choose whether or not to provide this information.

The qualitative data presented are based on the answers provided by the participants in the questionnaires, which were translated into English. These transcriptions faithfully reflect the original responses given during the course, without any alteration of content, and were used for qualitative data analysis.

The data were subsequently subjected to qualitative content analysis, based on Bardin’s framework [22,23]. Among Bardin’s techniques, categorial–thematic analysis [24] stands out, as it organizes the content into categories and themes according to the recurrence and relevance of meanings.

The analysis was performed using the following steps:Pre-analysis, with floating reading and the establishment of information about the topic, through exhaustiveness, representativeness, homogeneity, and relevance to the research objectivesExhaustive exploration of the material for coding, creation of context units, and categorizationInterpretation of the data and necessary inferences.

The study was conducted after approval from the ethics and research committee. Furthermore, all the ethical requirements necessary for conducting the research were properly respected. All participants provided informed consent before participating in the study. The study was conducted following the Declaration of Helsinki, and approved by the Ethics Committee of Federal University of Rio de Janeiro, Rio de Janeiro, Brazil (number 7.228.452 of 16 November 2024).

## 3. Results

### 3.1. Quantitative Data

In the study period 32 Last Aid Courses were offered in different settings in São Paulo, Rio de Janeiro and Online and 343 people attended.

Participants were divided according to the setting where they attended the course: people living in a favela, with courses offered in 3 favelas in Rio de Janeiro and São Paulo; university students from courses offered to medical, nursing and occupational therapy students; Community Health Workers (CHWs) who attended courses offered at their Primary Care Unit (PCU), and the General Public who attended face-to-face courses offered in public places or online. Table 1 shows the characteristics of each setting.

A total of 246 participants completed the questionnaire, resulting in a response rate of 72%.

The participants’ ages ranged from 19 to 82 years, with an average of 39.6 years. The self-declared gender characteristics of these participants are outlined in Table 2.

Table 3 shows the participants’ evaluation of the questions about learning new things during the course and a statement regarding whether they would recommend the course to others or not.

Table 4 presents the participants’ evaluation of the Last Aid Courses’ content on a five point scale ranking between very poor and very good.

When asked about their preference for the duration of the Last Aid Course, participants’ suggestions for the desired length of the Last Aid Course varied between 2 h and 2 days. Figure 1 shows the answers according to the different settings. 

### 3.2. Qualitative Data

Participants were asked about the most important message they received from the course. The responses were analyzed by qualitative content analysis and qualitative description. This process led to four main themes that are shown in Figure 2 and which are described in more detail below:


**Theme 1. Death as a natural part of life.**


This theme reflects the widespread acknowledgment that death, is an inevitable part of life that requires preparation. Although this fact is known by everyone, it is still a taboo to talk about it.

“Death is part of life, and there needs to be quality and humanity at this time as well.” *(General Public)*

It emphasizes the importance of facing death with understanding and quality care, regardless of whether one feels ready for it. It is important to be prepared to say farewell.

“The only certainty in life is death, we need to be prepared for this moment of farewell.” *(Favela)*

For many people it is paramount to be respected and to be able to die with dignity.

“The importance of dying with dignity.” *(University)*


**Theme 2. Importance of talking about death and dying.**


The responses of the informants reinforce the broad consensus across all groups that talking about death is not harmful but rather helps to reduce fear, fostering a more supportive environment for everyone involved.

“Talking about death does not hasten death, that is, it does not attract death.” *(PCU)*

“Talking about the process of dying is necessary and can be lighthearted.” *(University)*

Open communication about death, dying and grief allows individuals to express their thoughts, wishes, and fears, fostering an environment of respect.

“The importance of talking about death with our family, since death is inevitable. Maintaining a dialogue with loved ones is the best way to find out what they want, and also to make my wishes clear.” *(Favela)*


**Theme 3. Dying with dignity and respect for patients’ wishes.**


It is important to talk about death and dying to support dying people in the best way possible and to respect their autonomy and their wishes for end-of-life care as far as possible. The responses of the informants underline the importance of respecting the wishes of individuals at the end of life, ensuring a dignified death by honoring the patient’s decisions and preferences.

“Respect people’s wishes.” *(PCU)*

To know and understand the wishes of a person can also make decision-making and the whole dying process easier and might contribute to reduce conflicts.

“With understanding, I can make this process easier.” *(General Public)*

One main theme that was emphasized as a key takeaway was need for compassion and empathy during the dying process.

“Respect the patient’s decision because just a touch as a gesture of affection makes all the difference at this time.” *(University)*


**Theme 4. Education about palliative care.**


As death literacy is low in the public and most people have a lack of knowledge about serios illness, dying, death and grief education about palliative care is urgently needed. Participants across various groups expressed the need for greater education on palliative care.

“The importance of breaking the taboo around death.” *(University)*

“We need to talk about Palliative Care.” *(General Public)*

Many informants recognized the importance of discussing and understanding palliative care, which plays a crucial role in supporting both patients and their families during the end-of-life process.

“I learned about some ways to comfort, how to act and I learned about each religion and the importance of respecting the moment.” *(General Public)*

In summary the results from the qualitative data show that talking about death and dying should become more normal in the public and that talking about these themes is helpful. Communication about one’s own wishes for decision-making is important to respect autonomy and patient wishes for end-of-life care. Thus, the majority of participants highlight that education about death, dying, grief and palliative care is important and should be improved in the public space.

## 4. Discussion

The main results from the quantitative analysis of 32 courses that were held in various environments, attracting a total of 343 participants, were as follows:

Evaluations of the courses showed a high satisfaction rate, with 98.4% of participants expressing that they would recommend the course to others, and 98.8% indicating they had acquired new knowledge. Additionally, most attendees rated the course content as “very good” across all settings. This shows that the Last Aid Course is feasible and highly accepted by people in Brazil. The predominance of positive feedback can be attributed to participants’ genuine engagement with the topic and their recognition of the course’s relevance to both professional and personal practice.

Among all settings, the results from the analysis of the qualitative data highlight the importance of normalizing conversations about death, raising awareness of palliative care and maintaining dignity during the dying process.

The implementation of the Last Aid Course in Brazil showed positive results, indicating its potential to raise awareness about the topic, regardless of the context. The high satisfaction rates and the recognition of the course’s value among different participant groups, both in terms of local context and education, suggest that such initiatives can help break taboos surrounding death and improve the quality of end-of-life care. The findings of this study align with those from research conducted in other countries [11,13,18,25]. In a study conducted with 5469 participants of Last Aid Courses in Germany, Switzerland, and Austria, it was found that 99% found the course content easy to understand, and 99% would recommend the course to others. The overall course rating was ‘very good’ [25]. These findings are similar to our results from Brazil.

The online course assessed in this study also received favorable feedback from participants, reinforcing the findings of the previously mentioned study in Germany and Scotland [17,26]. This format allows participation from individuals who are unable to attend in person, including caregivers of severely ill patients, as well as those living in locations with limited accessibility to the course [17].

The Last Aid Course was delivered to 42 CHWs, who are key in mediating between the population and the healthcare system. In addition to disseminating essential health information, they facilitate the referral of community needs to the Family Health Strategy. This approach is designed to address territorial, cultural, and social diversity, aligning with the principles of the Brazilian Unified Health System (SUS) [27].

Participants, in general, consider the four-hour duration to be sufficient. However, the CHWs showed greater interest in attending a longer course, indicating the need for more education and training for this profession. This finding is similar to findings from Germany that people from healthcare professions would prefer a slightly longer course lasting one day [28,29].

The quantitative and qualitative results indicate that the current Last Aid Course curriculum seems to meet the growing demand for educational strategies in end-of-life care, as outlined in the National Palliative Care Policy [6]. Furthermore, it helps to fill an important gap in strengthening knowledge and practices in this type of care, ensuring a more integrated approach to the needs and demands of end-of-life care [30].

The Last Aid Course was initiated in Brazil in 2020. In addition to community training, efforts were also made to offer courses aimed at training facilitators for the Last Aid Course. This initiative contributes to the dissemination of knowledge on this topic, not only for the training of future facilitators but also for the development of individuals across different states in Brazil [16].

As in other countries, the dissemination of Last Aid Courses in Brazil has primarily occurred through word of mouth [13,16]. However, to expand its reach and impact, the course should be institutionally integrated into various settings.

Another promising initiative was the implementation of the Last Aid Course in favelas through partnerships with compassionate community projects [31]. The proposal contributes to enhancing the community’s ability to support its members by mobilizing volunteers and expanding the health support network, integrated with primary care services, aiming to reduce disparities in access to palliative care [10,32]. Currently, established compassionate communities exist in Rio de Janeiro [10,32], Goiânia, São Paulo, and Belo Horizonte, with additional initiatives under development [10,32,33]. Thus, the Last Aid Course can be seen as the educational basis for public palliative care education of people in compassionate communities [11].

The demand for palliative care at home will increase in the coming years [11,21]. Additionally, it is important to highlight that there are unique realities in Brazil. Some areas are dominated by drug trafficking or militias and lack basic sanitation. There are regions where access to healthcare facilities is hindered, especially for individuals with mobility challenges and those who are seriously ill [10,34]. The disparities in the country are vast in access to healthcare services [35]. The disparities in access to healthcare services across the country are vast, and the growing need for home-based care further emphasizes the urgent need for initiatives and strategies focused on community education, such as the Last Aid Course.

In Brazil, in addition to insufficient care at the end of life, as shown in the international rankings [1,5], we face a challenge in the training process of healthcare professionals. Education on Palliative Care only became mandatory in the medical curriculum in 2023 [36]. Data collected in 2021 showed that only 44 out of 315 (14%) medical schools registered with the Ministry of Education and Culture offered any type of education on Palliative Care [37,38]. In other areas of healthcare professional education, this topic is not mandatory. Thus, the Last Aid Course can contribute to educate the public about palliative care and to strengthen compassionate communities in their daily work. As first experiences with an extended Last Aid Course for healthcare professionals from Germany [29] are a future option could be to implement these courses in Brazil too.

Despite the universal and equitable principles of the Brazilian Unified Health System (SUS), disparities related to socioeconomic and regional factors continue to affect access to information and care. Populations in rural or peripheral urban areas face barriers such as low health literacy, fragmented services, and unequal distribution of professionals. Similar inequities persist in higher education, where socioeconomic background influences access to opportunities, despite affirmative action policies that have diversified public universities [32,34,35]. While this study did not explore these dimensions in depth, future research could address them more comprehensively.

Finally, the strong interest expressed by community health workers in longer and more detailed courses reflects both the novelty of the subject and existing educational gaps. As many community health workers do not have formal higher education, opportunities to deepen their understanding of palliative care may be perceived as particularly valuable. These insights should be considered in future evaluations of similar educational interventions to better tailor content and duration to participants’ diverse backgrounds and learning needs.

## 5. Limitations

One of the main limitations of the course is the relatively small amount of data collected from certain settings. For example, in the favela and for the CHWs, only 4 and 2 courses were held, respectively. Additionally, all the settings analyzed were located in the Southeast region of Brazil, which represents just one of the country’s five major regions.

It is also worth noting that not all participants completed the questionnaire (72%), and other opinions could be lacking. On the other hand, a response rate of 72% may provide a valid representation of the group of informants for the current study. 

Furthermore, this study did not include a pre-intervention assessment of participants’ baseline awareness or prior experience with palliative care. This limits the ability to measure knowledge changes directly attributable to the course. Future research may benefit from incorporating pre- and post-course evaluations to better assess the program’s educational impact and adaptation in the Brazilian context.

Finally, given the sensitive nature of end-of-life issues, social desirability bias could theoretically have influenced participants’ responses. This bias may lead individuals to provide answers perceived as socially acceptable rather than reflecting their true opinions. However, because the questionnaires were completed anonymously and without direct interaction with instructors, this risk was likely minimized.

## 6. Conclusions

The results of the current study indicate that using the standardized international Last Aid Course curriculum to discuss Palliative Care and end-of-life care can be effective in different settings and for participants with diverse backgrounds. This suggests that the Last Aid Course can serve as a simple and accessible tool to raise awareness of Palliative Care in different populations as shown for Brazil and many European countries.

The feedback from CHWs suggests that these professionals want more information about Palliative Care and would benefit from a Last Aid Course, and this may be an accessible tool to strengthen the National Palliative Care Policy, especially in Primary Healthcare. In the future the implementation of the Last Aid Course Professional might be an option for this special group.

As an innovative study, the findings show that the Last Aid Course model can be expanded and continued, offering significant potential for further implementation and impact in different regions and contexts of Brazil.

## Figures and Tables

**Figure 1 nursrep-15-00386-f001:**
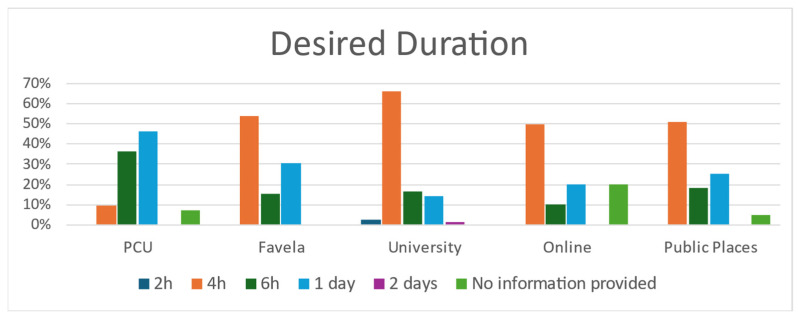
Participants’ suggestions for the desired duration of the Last Aid Course. PCU: Primary Care Unit.

**Figure 2 nursrep-15-00386-f002:**
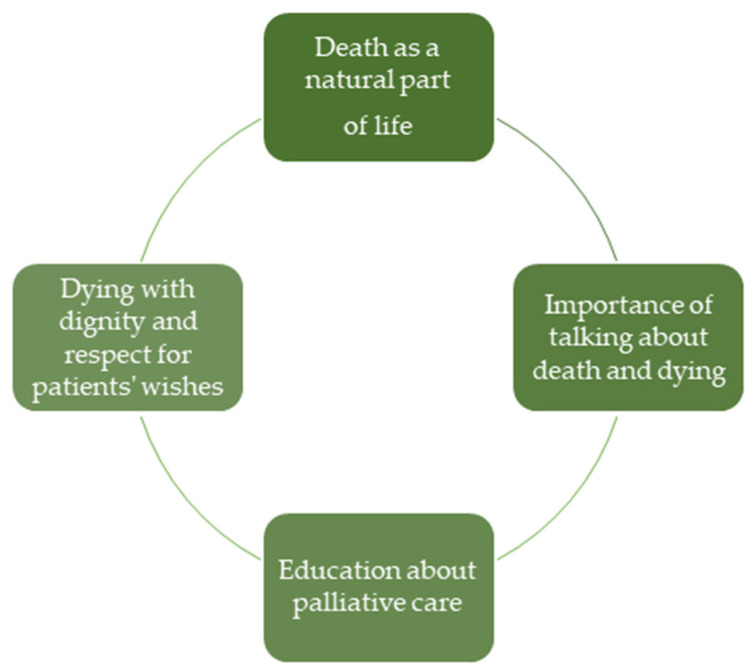
Themes from the qualitative data.

**Table 1 nursrep-15-00386-t001:** Settings and Participants of the Last Aid Courses.

Setting	nº of Courses	Participants
Favela	4 (12.5%)	32 (9.3%)
University	9 (28.1%)	105 (30.6%)
PCU ^1^	2 (6.25%)	42 (12.2%)
Public Place	10 (31.25%)	121 (35.3%)
Online	7 (21.9%)	43 (12.5%)
Total	32	343

^1^ PCU: Primary Care Unit.

**Table 2 nursrep-15-00386-t002:** Participants age and gender characteristics.

	Women	Men	No Information Provided	Participants
Favela	12 (92.3%)	1 (7.7%)	0 (0.0%)	13
University	69 (80.2%)	17 (19.8%)	0 (0.0%)	86
PCU	41 (100.0%)	0 (0.0%)	0 (0.0%)	41
Public Place	67 (77.9%)	17 (19.8%)	2 (2.3%)	86
Online	16 (80.0%)	4 (20.0%)	0 (0.0%)	20
Total	205 (83.3%)	39 (15.9%)	2 (0.8%)	246

PCU: Primary Care Unit.

**Table 3 nursrep-15-00386-t003:** Participants’ evaluation of the importance and relevance of the course.

	Yes	No	No Information Provided
I will recommend the course to others	242 (98.4%)	4 (1.6%)	0 (0.0%)
I learned new things	243 (98.8%)	2 (0.8%)	1 (0.4%)

**Table 4 nursrep-15-00386-t004:** Participants´ rating of the Last Aid course.

Setting	Very Poor (n, %)	Poor (n, %)	Neither Poor Nor Good (n, %)	Good (n, %)	Very Good (n, %)	No Information Provided (n, %)	Total Responses
PCU ^1^	1 (2.4%)	0	0	1 (2.4%)	38 (92.7%)	1 (2.4%)	41
Favela	0	0	0	0	13 (100%)	0	13
University	0	0	0	4 (4.7%)	81 (94.2%)	1 (1.2%)	86
Online	0	0	0	3 (15.0%)	17 (85.0%)	0	20
Public Places	0	0	0	11 (12.8%)	72 (83.7%)	3 (3.5%)	86

^1^ PCU: Primary Care Unit.

## Data Availability

The data presented in this study are available in part on request from the corresponding author. The data are not publicly available due to privacy restrictions.

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
