# Peer review of "First Experiences with Last Aid Courses as Tool for Public Palliative Care Education in Brazil"

_nursrep, 2025, doi:10.3390/nursrep15110386_

Round 1

Reviewer 1 Report

Comments and Suggestions for Authors

Dear Authors,

Thank you for the opportunity to review this valuable manuscript. I would like to offer some suggestions through the attached file. I hope the suggestions are helpful for strengthening your work.

Author Response

Dear Reviewer

Thank you very much for taking the time to review this manuscript.  Your comments were very helpful in improving the quality of our work.

Please find the detailed responses below.

Comments 1. There are some minor typographical errors in the manuscript. Please review and

correct them.

Response 1:

Thank you for your comment, we reviewed the errors in the text.

Comments 2. The rationale for using a mixed methods design should be clarified more explicitly

in relation to the study purpose. Since the program has been implemented since

2020, and if the intent was mainly to measure changes in knowledge and

awareness, it is not immediately clear why both quantitative and qualitative

approaches were necessary. Please explain in more detail why a mixed methods

approach was essential for this evaluation rather than relying solely on survey

data.

Response 2

The Last Aid Courses started 2020 in Brazil with the main focus being educational. The first years had only a very limited number of participants. After seeing an increase of courses this first Brazilian research-oriented evaluation started.

Being the first evaluation, the mixed methods approach was chosen to provide a richer picture, with more in-depth views of the participants. This information was added to the manuscript.

Comments 3. It would be helpful to clarify in the Introduction whether the study involved

participants from an already established LAC program or whether the program

was conducted specifically for this research purpose.

Response 3

Thank you for your questions. We will clarify this point in the introduction.

The LAC started with an educational focus and the research was started afterwards to explore participants’ perceptions, assess the course’s acceptance in the Brazilian context, and provide evidence to support its future expansion. This information was added to the manuscript

Comments 4. Please provide more details about participant recruitment. Was it conducted via

online advertisements, posters, or other methods? How was the sample size

determined? Did you check for participants’ prior experience with the LAC

program? If prior experience existed, this should be addressed as an exclusion

criterion or at least discussed. If all program participants were eligible, please

describe the admission criteria and recruitment process for the program itself.

Additionally, it would strengthen the paper to state the proportion of program

participants who agreed to take part in the study.

Response 4

Thank you for your questions

From March to Oct 2024, at the end of each course, everyone who attended a LAC was invited to evaluate the course. This information was added to the manuscript

In the study period 32 LACs were offered and 343 people attended. A total of 246 participants completed the questionnaire, resulting in a response rate of 72%.

Comments 5. Was there a pre-intervention assessment of participants’ baseline awareness or

perceptions? Such data would be important to understand whether post-program

outcomes varied depending on prior experiences or pre-existing knowledge about

palliative care.

Response 5

We appreciate this important observation. A pre-intervention assessment of participants’ baseline knowledge and perceptions was not conducted. The main objective of this study was to explore participants’ experiences and perceptions after attending the course. We acknowledge, however, that the absence of baseline data limits the ability to compare pre- and post-course changes in awareness. This limitation has been added to the discussion section

Comments 6. The stated study purpose is to explore whether the program contributes to

disseminating knowledge and awareness of palliative care. To align with this goal,

the Discussion should focus primarily on themes related to that outcome.

Statements about satisfaction, willingness to recommend, or perceptions of

program quality relate more to educational satisfaction rather than to awareness

or knowledge dissemination. If you wish to include those aspects, I suggest

revising the Introduction to broaden the study objectives accordingly.

I believe that addressing these points will enhance the clarity, methodological rigor, and

overall contribution of the manuscript. Thank you again for your important work in this

area.

Response 6

Thanks for pointing this out. We revised the study objectives accordingly

We hope we have answered your comments

Regards

The Authors

Reviewer 2 Report

Comments and Suggestions for Authors

This article is very well researched and clear in its organization and writing style. I especially appreciated that the author(s) included their full survey instrument beginning on page 9, and their research literature is well-documented and wide-ranging. I find their research method to be very strong overall and their study is convincing. The importance of end-of-life discussion and the Last Aid Courses in Brazil are clearly demonstrated in their study. In terms of the documentation of the various participants of the study and the locations in which the participants were able to access the LAC, I did wonder if the author(s) could have said more about the issues of racism and access to information about health care, as well as the questions of regional disparities that they mention on page 3 but did not go into more detail on. I do think the issues of access, regional disparities, racism, and economic status in Brazil are more complicated and I think the article could say more about these issues. (For example could there be more information included about which public places, which universities, where they were located when the questionnaire was completed.) But this is merely a suggestion for improvement and I do think the article deserves to be published. 

Author Response

Dear Reviewer

Thank you very much for taking the time to review this manuscript.  Your comments were very helpful in improving the quality of our work.

Please find the detailed responses below.

Comments 1

This article is very well researched and clear in its organization and writing style. I especially appreciated that the author(s) included their full survey instrument beginning on page 9, and their research literature is well-documented and wide-ranging. I find their research method to be very strong overall and their study is convincing. The importance of end-of-life discussion and the Last Aid Courses in Brazil are clearly demonstrated in their study. In terms of the documentation of the various participants of the study and the locations in which the participants were able to access the LAC, I did wonder if the author(s) could have said more about the issues of racism and access to information about health care, as well as the questions of regional disparities that they mention on page 3 but did not go into more detail on. I do think the issues of access, regional disparities, racism, and economic status in Brazil are more complicated and I think the article could say more about these issues. (For example could there be more information included about which public places, which universities, where they were located when the questionnaire was completed.) But this is merely a suggestion for improvement and I do think the article deserves to be published. 

Response 1

We appreciate this thoughtful comment. We agree that social and regional inequalities play an important role in shaping access to health information and care in Brazil. In response, we have added a paragraph in the Discussion section acknowledging that, although the Brazilian Unified Health System (SUS) is theoretically universal, persistent structural inequities continue to affect healthcare access and education.

Regarding participant institutions, the study involved both public and private universities. We recognize that private institutions in Brazil tend to concentrate students from higher socioeconomic backgrounds, while public universities have become more socially diverse through affirmative action policies. However, we did not collect socioeconomic or ethnicity data from participants, which we have now noted as a study limitation.

We hope we have answered your comments

Regards

The Authors

Reviewer 3 Report

Comments and Suggestions for Authors

The article addresses a highly relevant and timely topic within its context.

Technical Review Suggestions:

  • It is recommended to standardize the use of the terms “Last Aid Course” (LAC) and “Last Aid Courses” throughout the manuscript, avoiding variations such as “LACs” and “LAC”.
  • Replace the term “brakes” with “breaks” in the course duration description (p.2).
  • Review the formatting of references, ensuring completeness of DOIs and removal of unnecessary spaces.
  • Correct typographical errors, such as “ érios illness” → “serious illness”.
  • Ensure consistency in the use of punctuation (commas and periods) in lists and tables.

Abstract:

Strengths:

  • The manuscript employs clear and objective language.

Areas for Improvement:

  • Clarify the use of the mixed methods approach to better define the study typology.

Introduction:

Strengths:

  • The literature review is current and relevant to the topic.

Areas for Improvement:

  • Improve the cohesion between paragraphs to enhance argumentative flow.

Methodology:

Strengths:

  • The use of mixed methods (qualitative and quantitative) is appropriate and well-aligned with the study’s objectives.

Areas for Improvement:

  • Justify the choice of a descriptive qualitative approach. Although suitable, it would be beneficial to discuss why thematic or phenomenological analysis was not selected, and how this methodological decision may influence the findings.

Results:

Strengths:

  • Data are presented clearly and objectively, with tables that facilitate comprehension.

Areas for Improvement:

  • Present disaggregated data by age group, gender, and professional background of participants to enrich the analysis and allow for more nuanced interpretations.

Discussion:

Strengths:

  • The discussion effectively connects the findings with international literature and the Brazilian context.

Areas for Improvement:

  • Address potential social desirability bias, considering the sensitive nature of the topic and the course format, which may have influenced participants to respond more positively due to empathy with instructors or social pressure.
  • Analyze the absence of neutral or negative responses, as the predominance of positive evaluations may suggest selection bias or limitations in the evaluation instrument.
  • Explore in greater depth the reasons why healthcare professionals (e.g., Community Health Workers) express interest in longer courses, as this may reveal gaps in professional training that warrant further attention.

Conclusion:

Strengths:

  • The conclusion is clear, well-aligned with the study’s objectives, and reinforces the applicability of Last Aid Courses (LAC) in diverse Brazilian contexts. It appropriately highlights the relevance of LAC as a public education tool in palliative care and its potential contribution to the National Palliative Care Policy.

Areas for Improvement:

  • The conclusion could be strengthened by briefly outlining future research directions and strategies for long-term impact assessment.

Author Response

Dear Reviewer

Thank you very much for taking the time to review this manuscript.  Your comments were very helpful in improving the quality of our work.

Please find the detailed responses below.

Comments 1:It is recommended to standardize the use of the terms “Last Aid Course” (LAC) and “Last Aid Courses” throughout the manuscript, avoiding variations such as “LACs” and “LAC”.

Response 1: We revised the terms in the text

Comments 2: Replace the term “brakes” with “breaks” in the course duration description (p.2).

Response 2 We replaced the term in the text

Comments 3: Review the formatting of references, ensuring completeness of DOIs and removal of unnecessary spaces.

Response 3 We revised the formatting of references

Comments 4: Correct typographical errors, such as “ érios illness” → “serious illness”.

Response 4 We corrected the errors in the text

Comments 5:  Ensure consistency in the use of punctuation (commas and periods) in lists and tables.

Response 5 We revised the punctuation in lists and tables

Comments 6

Abstract:

Clarify the use of the mixed methods approach to better define the study typology.

Response 6: Being the first Brazilian course evaluation, the mixed methods approach was chosen to provide a richer picture, with more in depths views of the participants. This information was added to the manuscript.

Comments 7

Introduction:

Improve the cohesion between paragraphs to enhance argumentative flow.

Response 7

We revised the introduction and the argumentative flow

Comments 8

Methodology:

Justify the choice of a descriptive qualitative approach. Although suitable, it would be beneficial to discuss why thematic or phenomenological analysis was not selected, and how this methodological decision may influence the findings.

Response 8

We added more context in the methodology section

Comments 9

Results:

Present disaggregated data by age group, gender, and professional background of participants to enrich the analysis and allow for more nuanced interpretations.

Response 9:

Thank you for this suggestion, we will discuss this possibility in future evaluations.

Comments  10

Discussion

  • Address potential social desirability bias, considering the sensitive nature of the topic and the course format, which may have influenced participants to respond more positively due to empathy with instructors or social pressure.
  • Analyze the absence of neutral or negative responses, as the predominance of positive evaluations may suggest selection bias or limitations in the evaluation instrument.
  • Explore in greater depth the reasons why healthcare professionals (e.g., Community Health Workers) express interest in longer courses, as this may reveal gaps in professional training that warrant further attention.

Response 10

Thank you for your comments. We added this topic in the discussion section.

Comments  11

Conclusion

The conclusion could be strengthened by briefly outlining future research directions and strategies for long-term impact assessment.

Response 11

Thank you for your suggestion. The present study is a part of an ongoing larger research project which will evaluate long-term impacts.

In addition the manuscript has been proofread by a professional native speaker.

We hope we have answered your comments

Regards

The Authors

Round 2

Reviewer 1 Report

Comments and Suggestions for Authors

Dear Authors,

I have reviewed the revised manuscript and am pleased to see that you have addressed the feedback thoughtfully and made the necessary adjustments. Thank you for your diligent work.